# Characterization of Latin American migrants at risk for *Trypanosoma cruzi* infection in a non-endemic setting. Insights into initial evaluation of cardiac and digestive involvement

Pedro Laynez-Roldán[1][��]*, Irene Losada-Galván[1,2][��]*, Elizabeth Posada[1], Leonardo de la Torre Ávila[1], Aina Casellas[1], Sergi Sanz[1], Carme Subirà[1], Natalia Rodriguez-Valero[1], Daniel Camprubí-Ferrer[1], Isabel Vera[1], Montserrat Roldán[1], Edelweiss Aldasoro[3], Inés Oliveira-Souto[4], Antonia Calvo-Cano[5], Maria-Eugenia Valls[1,6], Míriam J. Álvarez-Martínez[1,6], Montserrat Gállego[1,7,8], Alba Abras[9], Cristina Ballart[1,7], José Muñoz[1], Joaquim Gascón[1,8], Maria-Jesus Pinazo[1,8]*

**1** Barcelona Institute for Global Health, ISGlobal-Hospital Clinic, Universitat de Barcelona, Barcelona, Spain, **2** Hospital Universitario 12 de Octubre, Madrid, Spain, **3** The International Foundation for Integrated Care (IFIC), Schiphol, The Netherlands, **4** Department of Infectious Diseases, Tropical Medicine & International Health Unit Vall d'Hebron-Drassanes, Vall d'Hebron University Hospital, Universitat Autònoma de Barcelona, PROSICS Barcelona, Barcelona, Spain, **5** Department of Infectious Pathology, Badajoz University Hospital, Badajoz, Spain, **6** Department of Microbiology, Hospital Clinic, Barcelona, Spain, **7** Parasitology section, Departament of Biology, Health and Environment, Faculty of Pharmacy and Food Science, Universitat de Barcelona, Barcelona, Spain, **8** Center for Network Biothecnological Research in Infectious Diseases (CIBERISCIII), Madridy, Spain, **9** Genetic Area, Department of Biology, Universitat de Girona, Campus Montilivi, Girona, Spain

☯ These authors contributed equally to this work.
* playnezroldan@gmail.com (PL-R); irene.losada@isglobal.org (IL-G); mpinazo@dndi.org (M-JP)

**Data Availability Statement:** The data present in this study are available upon request from the

## Abstract

### Background

*Trypanosoma cruzi* causes Chagas disease (CD), a potentially fatal disease characterized by cardiac disorders and digestive, neurological or mixed alterations. *T. cruzi* is transmitted to humans by the bite of triatomine vectors; both the parasite and disease are endemic in Latin America and the United States. In the last decades, population migration has changed the classic epidemiology of *T. cruzi*, contributing to its global spread to traditionally non-endemic countries. Screening is recommended for Latin American populations residing in non-endemic countries.

### Methods

The present study analyzes the epidemiological characteristics of 2,820 Latin American individuals who attended the International Health Service (IHS) of the Hospital Clinic de Barcelona between 2002 and 2019. The initial assessment of organ damage among positive cases of *T. cruzi* infection was analyzed, including the results of electrocardiogram (ECG), echocardiogram, barium enema and esophagogram.

"Catalan Open Research Area" with the identifier https://doi.org/10.34810/data223 (contact address: ubioesdm@isglobal.org), access to the data is restricted due to ethical reasons. The complete data dictionary (data_dictionary.pdf) listing the dataset variables, and the dataset details (readme. txt) are available without restriction. Additionally, the publication includes a supporting information file to facilitate reproducibility of Fig 1 (S1 Table).

**Funding:** ISGlobal work is supported by the Departament d'Universitats i Recerca de la Generalitat de Catalunya, Spain (AGAUR; 2017SGR00924) and by the Instituto de Salud Carlos III (ISCIII) RICET Network for Cooperative Research in Tropical Diseases (ISCIII; RD16/0027/0004 - PI1290) and FEDER. MJP research is supported by the Ministry of Health, Government of Catalonia (PERIS 2016-2010 SLT008/18/00132). The research of MJP and JG is supported by the Departament d'Universitats i Recerca de la Generalitat de Catalunya, Spain (AGAUR; grant 2014SGR26). ILG research is supported by "Centro de Excelencia Severo Ochoa 2019-2023" Program (CEX2018-000806-S), from the Spanish Ministry of Science and Innovation and State Research Agency. Specific funding has not been received for this study. The funders had no role in study design, data collection and analysis, decision to publish, or preparation of the manuscript.

**Competing interests:** The authors have declared that no competing interests exist.

## Results

Among all the screened individuals attending the clinic, 2,441 (86.6%) were born in Bolivia and 1,993 (70.7%) were female. Of individuals, 1,517 (81.5%) reported previous exposure to the vector, which is a strong risk factor associated with *T. cruzi* infection; 1,382 individuals were positive for *T. cruzi* infection. The first evaluation of individuals with confirmed *T. cruzi* infection, showed 148 (17.1%) individuals with Chagasic cardiomyopathy, the main diagnostic method being an ECG and the right bundle branch block (RBBB) for the most frequent disorder; 16 (10.8%) individuals had a normal ECG and were diagnosed of Chagasic cardiomyopathy by echocardiogram.

## Conclusions

We still observe many Latin American individuals who were at risk of *T. cruzi* infection in highly endemic areas in their countries of origin, and who have not been previously tested for *T. cruzi* infection. In fact, even in Spain, a country with one of the highest proportion of diagnosis of Latin American populations, *T. cruzi* infection remains underdiagnosed. The screening of Latin American populations presenting with a similar profile as reported here should be promoted. ECG is considered necessary to assess Chagasic cardiomyopathy in positive individuals, but echocardiograms should also be considered as a diagnostic approach given that it can detect cardiac abnormalities when the ECG is normal.

## Author summary

*Trypanosoma cruzi* is a protozoan infection that can be transmitted to humans by triatomine insects, endemic from 21 Latin American countries. It can also be transmitted vertically (mother to child) and by blood transfusions, among other less frequent methods. *T. cruzi* infection is called Chagas Disease (CD) when causing organ damage, such as Chagasic cardiomyopathy (CC) and digestive involvement in 30–40% of cases. The large migration flows from Latin America to Europe have globalized the distribution of *T. cruzi* infection. Thus, screening is recommended for Latin Americans living in non-endemic countries. We have retrospectively analyzed the epidemiologic characteristics of individuals from endemic countries screened for *T. cruzi* infection in Barcelona over 17 years, revealing a great number of working-age women coming from highly endemic areas for *T. cruzi* infection, which reflect the migration movements of the last decades and help us to focus the screening and health promotion programs. We have also analyzed the initial organ damage assessment, which revealed a great proportion of right bundle branch block and left anterior fascicular block, considered typical CD lesions (although unspecific). We also found an important proportion of patients with an altered echocardiogram but having a normal ECG, which reinforces the echocardiogram as an essential test for the assessment of CC.

## Introduction

American trypanosomiasis was first described by Carlos Chagas in 1909 [1]. Currently, we refer to *Trypanosoma cruzi* infection when no evidence of organ damage is present, while Chagas disease (CD) is the term used for the illness that results from *T. cruzi* infection. Up to 21

Latin American countries and the United States are considered endemic for *T. cruzi* infection and it is estimated to affect 6–7 million people [2]. Of all endemic countries, Bolivia has by far the highest reported prevalence, currently estimated at 6.1% [2]. Bites by triatomine vectors still represent the main route of transmission of *T. cruzi* to humans, but transmission is also possible orally (i.e., by ingestion of contaminated juices and foods) [3], congenitally from mother to child, by blood transfusion or organ donations [3–8].

In less than 5% of cases, individuals develop an acute symptomatic form of the disease [4]. After the acute phase is over, and if the infection remains untreated, *T. cruzi* progressively sets in the tissues with a high tropism for the heart and digestive tract. However, 60–70% of infected patients will not ever develop clinically significant organ damage [4]. Cardiac involvement is the most common organ damage, known as Chagasic cardiomyopathy (CC), which accounts for 25–30% of patients suffering from *T. cruzi* infection. Conduction disorders and dilatation of cardiac structures are typical manifestations, whose clinical spectrum comprise from asymptomatic to fatal outcomes [4]. Digestive involvement of CD (DCD) is less common (10–20%) and includes dilatation and dysfunction of digestive structures -mainly esophagus and colon [4].

Serological tests are the technique of choice for diagnosis of chronic *T. cruzi* infection [5]. Screening for cardiac involvement is indicated even in asymptomatic patients. The 12 lead ECG is the recommended test to detect conduction disorders and arrhythmias and should be offered to all patients [6,7]. Structural cardiomyopathy is classically assessed by thorax radiography, but last consensus recommends echocardiogram as it has more sensitivity and specificity [7–9]. Barium enema and esophagogram are recommended to assess digestive abnormalities due to CD but should only be performed when digestive symptoms are present [10].

In the last two decades, the global distribution of *T. cruzi* infection has changed due to population migration from endemic Latin America countries to Europe and other non-endemic settings [11]. In the early 2000s there was a large migration of Bolivian population towards Europe, mainly to large urban areas in Spain, due to the economic crisis and immigration restrictions in place in the USA [12].

A 2015 metanalysis estimated an average prevalence of *T. cruzi* infection of 4.2% in Latin American migrants living in European countries and, currently, Spain is the European country with the highest estimated prevalence [13]. Moreover, new cases of *T. cruzi* infection can occur in these traditionally non-endemic countries by non-vector-borne transmission. Thus, although screening for *T. cruzi* in populations originating from high endemic areas in Latin America should recommended in non-endemic countries, most European countries do not include this approach in their national health policies [14]. In Spain, there is no common legislation or initiative for *T. cruzi* screening; yet, in Catalonia an official program to detect and treat congenital *T. cruzi* infection has been implemented since 2010 [15].

In the last twenty years, several descriptions about clinical and epidemiological features of migrants at risk of *T. cruzi* infection have been published, but usually focused on blood donors or pregnant women, and the sample sizes of such studies were not large enough to describe the epidemiological profile of this population [16–20]. Here we present the clinical and epidemiological profile of a large sample of Latin American migrants attending the International Health Service (IHS) of the Hospital Clínic of Barcelona, Spain, which is the reference center for tropical diseases in the region.

## Materials and methods

### Ethics statement

All participants provided written informed consent for screening and data collection, in compliance with all bioethical regulations. All participants were over 16 years of age. Thus, none of

the participants required consent by representation, following applicable legislation [21]. This study has been approved by the Ethics Committee named "Comité de Ética de la Investigación con medicamentos del Hospital Clínic de Barcelona" (reference HCB/2021/0174).

## Design, setting, and recruitment

Barcelona has been one of the urban areas in Spain where a large Bolivian migrant population settled in the early 2000s [12], and the IHS of Hospital Clinic de Barcelona and Barcelona Institute of Global Health (ISGlobal) has a 17-year of experience in the clinical management and follow-up of this population. We conducted a descriptive retrospective study of 2,820 individuals originating from endemic Latin American countries considered *T. cruzi* endemic areas. Study subjects attended the medical consultation outpatient clinic of the IHS from March 2002 to March 2019. Subjects came to IHS (i) voluntarily, (ii) for screening after a positive result of a relative, (iii) upon advice by family or friends, (iv) referred by another doctor and (v) following information, education, and communication (IEC) community outreach.

The following countries were considered *T. cruzi* endemic areas: Argentina, Belize, Bolivia, Brazil, Chile, Colombia, Costa Rica, Ecuador, El Salvador, Guatemala, French Guyana, Guyana, Honduras, Mexico, Nicaragua, Panama, Paraguay, Peru, Suriname, Uruguay and Venezuela. Exclusion criteria for individuals to be included in the cohort were: a) individuals born in a non-endemic country; b) individuals under 16-years-old; c) inconclusive serology test; d) individuals who had not signed the informed consent.

Epidemiological and clinical data were collected from electronic medical records and questionnaires over the study period and registered in a database [22], and published in a data repository [23]. Collected variables included age, sex, country and department of origin, date of arrival to Spain, risk factors for *T. cruzi* infection and *T. cruzi* serology result. Also, the results of complementary tests performed by protocol were collected: ECG and echocardiogram, and barium enema and esophagogram when performed. Patients were then classified as follows: *T. cruzi* infection with no evidence of organ damage, CC and DCD. Only the first evaluation was included in this study to ascertain the phase of the infection at diagnosis.

## Diagnosis of *T. cruzi* infection and organ damage

Diagnosis of *T. cruzi* infection was performed by two serological tests using different antigens, two different enzyme-linked immunosorbent assays (ELISA) methods, ORTHO Trypanosoma cruzi ELISA Test System (Johnson and Johnson, High Wycombe, United Kingdom) and Bioelisa Chagas (Biokit, Lliçà d' Amunt, Spain) during the period (2002–2012) and chemiluminescent microparticle assay (CMIA ARCHITECT Chagas Abbot) and ELISA (Chagas-ELISA Vircell, Spain) in the period (2013–2019). Indirect Immunofluorescence (IFA Vircell, Spain) and western blot were used in case of discordant serologic results [24].

Patients with two positive tests were classified as suffering from *T. cruzi* infection. CC was assessed through clinical history, physical exam, a 12-lead ECG, a thorax radiography, and echocardiogram. The following ECG abnormalities were considered as CC: bradycardia less than 50 beats per minute, auriculoventricular block (AVB), any type of bundle branch block or hemiblock, frequent ventricular premature beats, tachyarrhythmia of any origin, ST-T changes, electrically inactive areas, and pacemaker rhythm as it implies a conduction disorder [18]. The following echocardiogram abnormalities were considered as CC: any ventricular or atrial dilatation, contractility disorders, wall aneurisms and diastolic dysfunction [25,26]. Barium enema or esophagogram were performed when lower or upper digestive symptoms (respectively) compatible with CD were present. Colon dilatation in the barium enema was consider as DCD. Megacolon was defined as a diameter of more than 6.5 centimetres in

descending colon, more than 8 centimetres in ascending colon and more than 12 centimetres in cecum [27,28]. Any gastroesophageal dysfunction or esophageal dilatation was also considered as DCD. Only the first evaluation for *T. cruzi* infection for each patient was included in this study.

## Statistical analysis

Data were described as frequencies, means (standard deviation, SD) or medians (interquartile range, IQR) for qualitative and quantitative variables, respectively. Chi-squared and Wilcoxon Rank Sum test were performed to assess differences between groups for qualitative and quantitative variables, respectively. Venn diagrams were used to summarize the relationship between *T. cruzi* infection risk factors. To quantify the relationship between patient characteristics and *T. cruzi* infection, odds ratios (OR) were estimated by means of logistic regression models. The significance level was set at 0.05. The analysis was carried out using Stata17 (@StataCorp. 2019. Stata Statistical Software: Release 16. College Station, TX: StataCorp LLC). For graphical representation R Foundation for Statistical Computing (@R Core Team. 2019. R: A Language and Environment for Statistical Computing. R Foundation for Statistical Computing. Vienna, Austria), with the package venneuler (Wilkinson, L. 2011. Venneuler: Venn and Euler Diagrams. R package version 1.1–0), were employed.

## Results

### General demographic and epidemiological variables

A total of 2,820 individuals born in *T. cruzi* endemic countries were included in the 17 years study period. The number of patients tested for *T. cruzi* infection per year progressively increased until 2009, when there was a peak of 356 individuals. The mean age was 36.8 years (SD 11.0). The median (IQR) of the duration of residence in Spain until the first visit to the IHS was six years (3.00–10.00) [2,813]. Most individuals came from Bolivia (Table 1), and 93.2% (n = 2,275/2,441) of these where from Cochabamba, Santa Cruz, and Chuquisaca Departments, all of which are considered highly endemic areas for *T. cruzi* transmission (Table 1). The overlap between reported risk factors is represented in the Fig 1. The greatest overlap (n = 987/1,889 individuals, 52.2%) was between the three risk factors related to the vector (rural area, adobe house, and exposure to the vector), while 101 individuals (N = 1,889; 5.35%) reported also having a mother with *T. cruzi* infection, and 114 (N = 1,889; 6.0%) had been residing in a rural endemic area and in an adobe house.

### Seropositivity and its associated factors

Among 2,820 participants, 1,382 participants were diagnosed of *T. cruzi* infection based on serology (Fig 2). Variables associated with *T. cruzi* infection were analyzed and are summarized in Table 2. Proportion of *T. cruzi* infection by country is detailed in Table 3.

### Organ damage in patients with *T. cruzi* infection: Chronic Chagas disease

Out of 1,382 patients with a confirmed positive result for *T. cruzi*, the first ECG of 662 patients was recorded in the study database. Among them, 79.0% (n = 523/662) had a normal result, while 21.0% (n = 139/662) had an altered result and 22 of them were found to have more than one disorder in the ECG. Among the 178 patients with an echocardiogram performed and recorded in the database, a total of 122 patients (68.5%) had a normal result, while 56 of them (31.5%) had an altered result. Cardiac dilatation was the most frequent disorder (n = 19; 10.67%): 12 of them with ventricular dilatation (6.7%), seven patients with atrial dilatation

**Table 1. General demographic and epidemiological variables of participants (N = 2,820).**

| Variable | n | Percentage (%) |
|---|---|---|
| **Female sex (N = 2,820)** | 1,993 | 70.7 |
| **Age range[1] (N = 2,820)** | | |
| <18 | 93 | 3.3 |
| 18–34 | 1,175 | 41.7 |
| 35–49 | 1,173 | 41.6 |
| 50–65 | 356 | 12.6 |
| <65 | 23 | 0.8 |
| **Country of origin (N = 2,820)** | | |
| Bolivia | 2,441 | 86.6 |
| *Cochabamba department* | *991* | *40.8* |
| *Santa Cruz department* | *783* | *32.2* |
| *Chuquisaca department* | *501* | *21.0* |
| *Other departments* | *155* | *6.4* |
| *Unknown* | *9* | *0.4* |
| Ecuador | 80 | 2.8 |
| Argentina | 73 | 2.6 |
| Colombia | 51 | 1.8 |
| Peru | 46 | 1.6 |
| Paraguay | 34 | 1.2 |
| Brazil | 24 | 0.9 |
| Honduras | 19 | 0.7 |
| Venezuela | 16 | 0.6 |
| El Salvador | 15 | 0.5 |
| Chile | 6 | 0.2 |
| Mexico | 6 | 0.2 |
| Uruguay | 4 | 0.1 |
| Guatemala | 3 | 0.1 |
| Nicaragua | 2 | 0.1 |
| **Referral (N = 1,893)** | | |
| Advised by family member or friend | 1,081 | 57.1 |
| Referred by a doctor[2] | 495 | 26.2 |
| IEC[3] | 129 | 6.8 |
| Own initiative | 37 | 2.0 |
| Other | 150 | 7.9 |
| **Vectorial risk factors (N = 1,889)** | | |
| Rural area | 1,517 | 80.3 |
| Adobe house | 1,506 | 79.7 |
| Contact with vector | 1,517 | 81.5 |
| **Non-vectorial risk factors** | | |
| Blood transfusion receptor (N = 2,773) | 188 | 6.7 |
| Mother with *T. cruzi* infection (N = 2,746) | 232 | 8.5 |
| **Report a previous test before this assessment (N = 1,916)** | | |
| Yes | 657 | 34.3 |
| No | 1,259 | 65.7 |
| **Positive serology (N = 2,820)** | 1,382 | 49.0 |

1. Age range is expressed in years.

2. Among individuals referred by a doctor, 73.5% (n = 364/495) were referred by a primary care doctor.

3. IEC: Information, education, and communication.

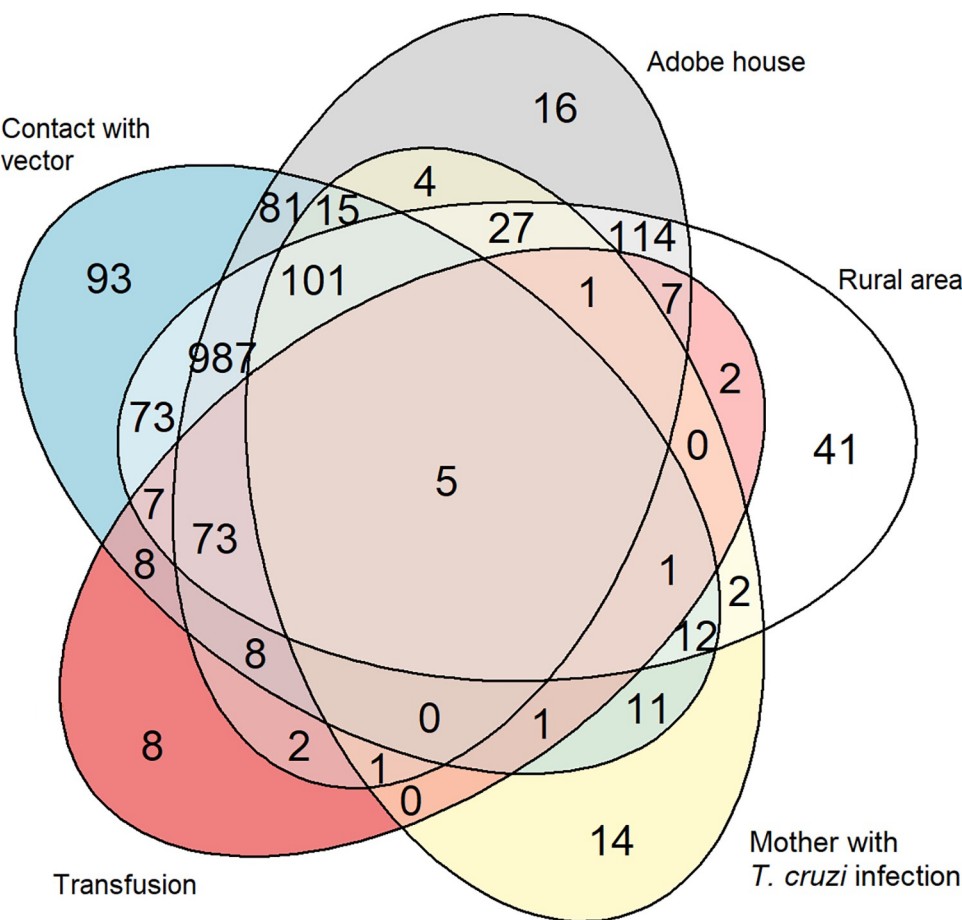

**Fig 1. Venn diagram showing the overlap between risk factors of *T. cruzi* infection. Venn diagram showing the overlap between risk factors of *T. cruzi* infection recorded in the database.** Each oval represents a risk factor. The numbers represent how many individuals have the corresponding risk factor to the area where it is located in the diagram.

(3.9%); two of them having a four-chamber dilatation (1.1%). Other findings, not considered to be CD related were: 14 patients with mild valve insufficiency, four of them with more than one valvopathy (11 cases of tricuspid insufficiency, six of mitral insufficiency, and one case of aortic insufficiency), and six had left ventricular hypertrophy. One patient had a severe pericardial effusion due to a non-chagasic etiology. Out of 153 patients with registered ejection fraction, six had a left ventricle ejection fraction (LVEF) of less than 35%, two patients had a LVEF between 35 and 45% and 145 had more than 45%. Main findings in the ECG and echocardiogram are summarized in Table 4.

In sum, 148 participants were diagnosed with CC. Of them, 16 patients (10.8%) had a normal ECG but with altered echocardiogram, nine of whom due to diastolic dysfunction (56.3% of them).

An esophagogram was recorded in 66 patients. Of these, 89.4% had a normal test (n = 59) and seven patients had a gastroesophageal dysfunction. Barium enema was registered in 71 patients, of whom 30 (42.3%) had a normal test. Dolichocolon was present in 29 patients, colon dilatation in 11 patients, including two who met the megacolon criteria. Cardiac involvement was present in patients from Argentina (n = 2), Bolivia (n = 138), Brazil (n = 3), Colombia (n = 2), Ecuador (n = 1), Paraguay (n = 1) and Venezuela (n = 1), while it was not present

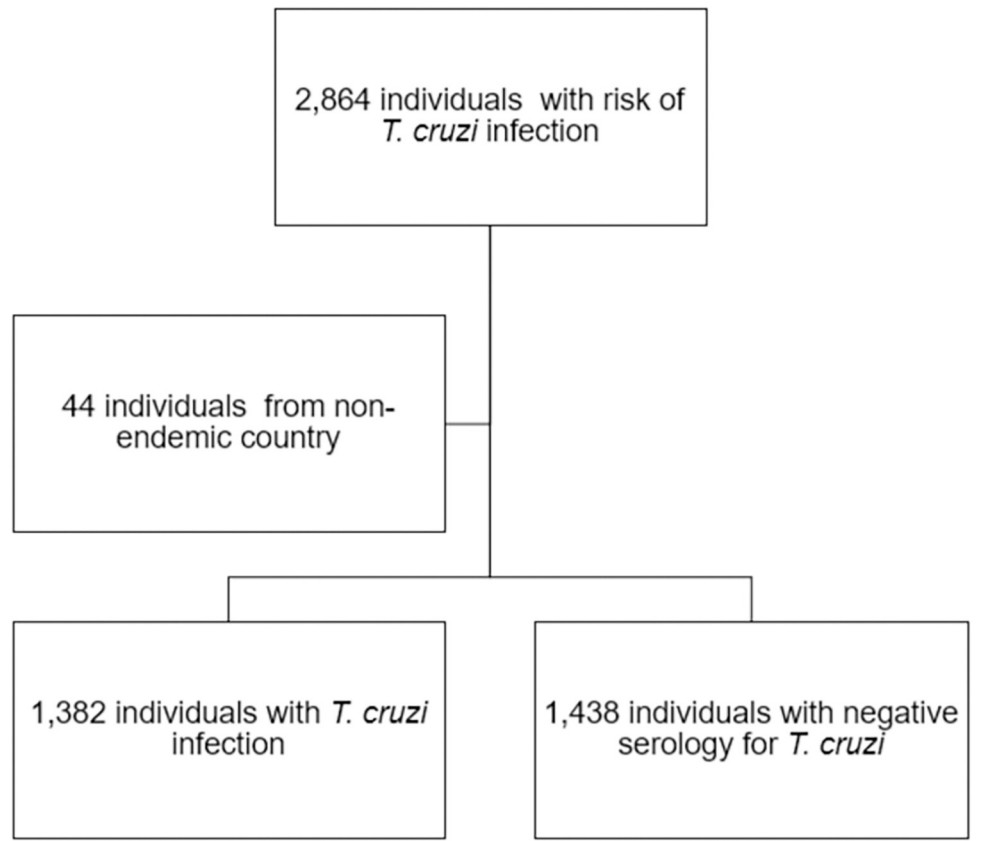

**Fig 2. Flowchart of screened individuals.** Flowchart of screened participants of this study. Diagnosis of *T. cruzi* infection was based on the serology.

in patients from other countries. Patients with digestive involvement were from Argentina (n = 1) and Bolivia (n = 22), while it was not present in patients from other countries. Regarding the frequency of cardiac or digestive involvement in the first evaluation, 148 patients (17.1%) were registered to have CC, while 23 patients (2.6%) were recorded as DCD and 703 patients (80.8%) did not have any evidence of organ damage. Due to missing data, 512 patients were excluded of this analysis.

## Discussion

Several studies performed in non-endemic countries have assessed the clinical and epidemiological profile of people at risk for *T. cruzi* infection [13,17–19,29,30]. However, most of these studies, especially in the USA [31–33], had either small sample sizes or focused on specific populations (e.g., blood donors or pregnant women). Furthermore, prevalence of *T. cruzi* infection in these studies was very low because most of the migrants who settled in North America originated from countries with low *T. cruzi* prevalence [31–33]. To our knowledge, this study describes one of the largest sample of patients at risk of *T. cruzi* infection in a non-endemic setting.

Trends in the influx of medical consultations are determined by Bolivian population migratory flows, since this population accounts for 86.6% of the participants. It is worth noting that patients from our study spent a median of six years in Spain before screening for *T. cruzi* infection was performed, which is longer than observed in other previous European studies [29,34].

**Table 2. Variables associated with *T. cruzi* infection using a multivariable model.**

| Variable | | Positive serology n/N (%) | p-value[1] | Adjusted OR (95% CI) | p-value[2] |
|---|---|---|---|---|---|
| **Sex** | **Male** | 236 / 566 (41.7) | 0.0091 | 1 | 0.0454 |
| | **Female** | 638 / 1323 (48.2) | | 1.24 (1.00, 1.54) | |
| **Age in years** | **<18** | 3 / 65 (4.6) | <0.0001 | 1 | <0.0001 |
| | **18–34** | 313 / 733 (42.7) | | 10.07 (3.08, 32.90) | |
| | **35–49** | 401 / 809 (49.6) | | 12.84 (3.93, 41.94) | |
| | **50–65** | 150 / 267 (56.2) | | 19.50 (5.86, 64.93) | |
| | **>65** | 7 / 15 (46.7) | | 15.34 (3.16, 74.56) | |
| **Born in Bolivia** | **Yes** | 833/1,667 (50.0) | <0.0001 | 3.12 (2.14, 546) | |
| | **No** | 41/181 (18.5) | | 1 | <0.0001 |
| **Resided in rural area** | **Yes** | 768/1,517 (50.6) | <0.0001 | 1.56 (1.15, 2.11) | 0.0039 |
| | **No** | 106/372 (28.5) | | 1 | |
| **Resided in adobe house** | **Yes** | 775/1,506 (51.5) | <0.0001 | 1.73 (1.27, 2.35) | 0.0004 |
| | **No** | 99/383 (25.9) | | 1 | |
| **Contact with vector** | **Yes** | 795/1,540 (51.6) | <0.0001 | 1.97 (1.46, 2.65) | <0.0001 |
| | **No** | 79/349 (22.6) | | 1 | |
| | **Total** | 874/1,889 (46.3) | | | |

1: Chi-squared test

2: Wald test

OR: Odds ratio, CI: confidence interval

Note that we are summarizing only those observations with complete data for the multivariable model (N = 1,889)

This delay may be explained by barriers to healthcare, such as the population's legal immigration and residential status, poor socio-economic conditions, and cultural barriers. It is estimated that a significant number of people living in Spain with *T. cruzi* infection are unaware of their condition. In fact, the index of underdiagnosed cases of *T. cruzi* infection is estimated to be of 71% [35] and, indeed, most of our participants had never been tested for *T. cruzi*

**Table 3. Frequency of *T. cruzi* infection according to country of origin.**

| Country of origin | Positive for *T. cruzi* test |
|---|---|
| **Bolivia (n = 2,441)** | 1,304 (53.4%) |
| **Paraguay (n = 34)** | 17 (50.0%) |
| **Argentina (n = 73)** | 31 (42.5%) |
| **Brazil (n = 24)** | 8 (33.3%) |
| **Chile (n = 6)** | 2 (33.3%) |
| **Venezuela (n = 16)** | 4 (25.0%) |
| **El Salvador (n = 15)** | 3 (20.0%) |
| **Mexico (n = 6)** | 1 (16.7%) |
| **Honduras (n = 19)** | 2 (10.5%) |
| **Ecuador (n = 80)** | 6 (7.5%) |
| **Peru (n = 46)** | 3 (6.5%) |
| **Colombia (n = 51)** | 3 (5.9%) |
| **Guatemala (n = 3)** | 0 (0.0%) |
| **Nicaragua (n = 2)** | 0 (0.0%) |
| **Uruguay (n = 4)** | 0 (0.0%) |

**Table 4. Main ECG and echocardiogram findings in patients with *T. cruzi* infection.**

| ECG findings | Frequency | Proportion among patients with altered ECG (n = 139) | Proportion among patients with ECG performed (n = 662) |
|---|---|---|---|
| Normal ECG | 523 | - | 79.0% |
| RBBB | 64 | 46.0% | 9.7% |
| LAFB | 40 | 28.8% | 6.0% |
| Sinus bradycardia | 17 | 12.2% | 2.6% |
| RBBB + LAFB | 13 | 9.4% | 2.0% |
| AVB | 11 | 7.9% | 1.7% |
| Nonspecific ST-T changes | 7 | 5.0% | 1.1% |
| VES | 6 | 4.3% | 0.9% |
| Supraventricular arrythms | 6 | 4.3% | 0.9% |
| Pacemaker | 5 | 3.6% | 0.8% |
| Electrically inactive area | 2 | 1.4% | 0.3% |
| Low voltage | 1 | 0.7% | 0.2% |
| LBBB | 1 | 0.7% | 0.2% |
| Other disorders | 5 | 3.6% | 0.8% |
| **Echocardiogram findings** | **Frequency** | **Proportion among patients with altered echocardiogram (n = 56)** | **Proportion among patients with echocardiogram performed (n = 178)** |
| Normal echocardiogram | 122 | - | 68.6% |
| Mild valve insufficiency | 14 | 25.1% | 7.9% |
| Ventricular dilatation | 12 | 21.4% | 6.7% |
| Diastolic dysfunction | 10 | 17.9% | 5.6% |
| Atrial dilatation | 7 | 12.5% | 3.9% |
| Contractility disorders | 7 | 12.5% | 3.9% |
| Ventricular hypertrophy | 5 | 8.9% | 2.8% |
| Wall aneurysm | 4 | 7.1% | 2.3% |
| Global hypokinesia | 4 | 7.1% | 2.3% |
| Other disorders | 4 | 7.1% | 2.% |

RBBB. Right bundle branch block. LAFB: Left anterior fascicular block. AVB: Atrioventricular block. VES: Ventricular extrasystoles. LBBB: Left bundle branch block. LPFB: Left posterior fascicular block.

infection prior to our assessment. The great predominance of women and working-age people from endemic countries reflects the demographic profile of Latin-American migrants who came to Europe in the early 2000's, as job offers were usually as domestic workers, which was socially attributed to women [12].

Health promotion campaigns among the Bolivian community were decisive in defining the attendance trends in the medical attention seeking in the IHS of Hospital Clinic de Barcelona, as shown by several health promotion programs directed to communities from *T. cruzi* endemic countries [36]. Most patients came because they were advised by a family member or a friend, which reveals the great importance of the social capital in the knowledge and care of the disease.

This study presents higher proportion of *T. cruzi* infection among participants than other European studies [13]. However, a selection bias was present because the participants were attending a referral center to which sometimes they were directed after a positive test. For this reason, proportion of seropositivity from this study should never be used to estimate the prevalence of *T. cruzi* infection in the general Latin American population living in Spain, which was already estimated to be of 2.1% in a recent study [35]. Nevertheless, we must consider that

Latin Americans in Barcelona come from highly endemic areas compared with those in other European cities [37]. In fact, most individuals were from the Bolivian departments of Cochabamba, Santa Cruz, and Chuquisaca, which are hyperendemic areas of *T. cruzi* infection [5].

Factors of potential vector exposure (adobe house, rural area, to have been in contact with the vector) were present in most patients, but there was a significant overlap among all the risk factors. Thus, many participants who had received a transfusion in an endemic country or whose mother had *T. cruzi* infection, have also had potential vector exposure.

Congenital and transfusion risks account for the possibility of new cases in Europe and justifies the implementation of screening programs for these risk groups as a priority in health public policies [14]. Moreover, screening for *T. cruzi* infection in all Latin American adults living in Europe has been shown to be cost-effective [37]. Additionally, young people infected with *T. cruzi* on an early chronic phase of *T. cruzi* infection seems to be the most benefited from an early diagnosis and anti-trypanosoma therapy [38,39]. Even more so in women of childbearing age, since treatment before pregnancy dramatically reduces the chances of congenital transmission [40,41].

Results from ECG assessment show RBBB as the most frequent and with the strongest association with *T. cruzi* infection, consistent with previously published studies [42]. LAFB was more frequent than sinus bradycardia, contrary to what was observed in previous studies [20,43]. Although the sensitivity of the ECG may be limited because it cannot evaluate function and morphology, it is an accessible and low-cost test capable of detecting potentially serious abnormalities. Indeed, more than 10% of patients with diagnosis of cardiac involvement had a normal ECG result and were diagnosed by echocardiogram. Similar data observing altered echocardiogram with normal ECG have been previously published [44,45]. Therefore, echocardiogram is considered a necessary test for this evaluation by most authors and guidelines [7–9], our findings pointing in the same direction. Typically, wall dilatation has been considered the main echocardiogram disorder in patients with *T. cruzi* infection, but diastolic dysfunction is suggested to be an earlier alteration of CC [26,46]. Our study shows diastolic dysfunction as the second most frequent disorder among patients with *T. cruzi* infection. Although this frequency is not very high compared with that of general population [47], its presence may be considered of Chagasic etiology, especially taking into account the young age of our patients. Presence of isolated mild tricuspid or mitral insufficiency without dilated ring should not be interpreted as a CC, as several studies have demonstrated a high prevalence (up to 80%) of mild regurgitation in healthy subjects [48] and, to our knowledge, there are no studies which associate it with disease progression. Regarding digestive involvement, dolichocolon was very frequent in patients with a barium enema performed. In agreement with previous work [10,49,50], dolichocolon was not considered as a digestive involvement for CD, thus the observed frequency of digestive involvement was relatively low. Furthermore, digestive involvement was only present in patients from southern countries of Latin America (Bolivia and Argentina). This would be consistent with the notion of tissue specific tropism of different strains of *T. cruzi* with characteristic geographic distribution [51], but the validity of this observation cannot be concluded since practically all the patients come from those same countries.

More than 80% of patients were not found to have CC or DCD in the first evaluation. This form of the disease is classically known as "indeterminate form" of chronic *T. cruzi* infection [4] or rather "chronic phase without demonstrable pathology" [52]. Most patients were young, so a number of them will develop some form of organ damage in the following years at a rate of 1–2% per year [53]. Hence, age could explain the lower prevalence of CC and DCD in studies from non-endemic countries compared with those from endemic countries [29]. The lack of clinical and biological markers of progression limits a more accurate follow-up in this big

group of patients. The future availability of such tools should be highly relevant for patients' clinical management and healthcare routes [54].

One limitation of this study is that comorbidities were not specifically addressed. We acknowledge that screening for comorbidities will become increasingly important with the aging of Latin American migrants, but we posit that the impact on the overall findings of this study would probably be small due to the mean and median age of the participants included in our study. Also recall bias could be present in the answers to the epidemiological question-naires. As the participants were recruited in a regional reference center a selection bias was present. Observer bias could also be present as electrocardiogram is an observer-dependent test, and the questionnaires were filled in by different physicians. This bias was probably reduced as diagnostic criteria of electrocardiographic disorders were established in detail. We were able to include the status of the patient at the moment of the screening and at the moment of the initial diagnosis of *T. cruzi* infection, but not their follow-up, that is why the number of ECG and echocardiogram recorded for the analyses was little compared with total sample size. Despite this discordance, the absolute number of tests registered and analyzed was also high and allowed us to extract enough data for the analysis and discussion. It would be interesting to analyze the longitudinal evolution of this cohort in terms of the prognostic value of the alterations found in this first evaluation.

## Conclusions

Currently, *T. cruzi* infection is distributed throughout Europe due to migratory flows from Latin America of the last two decades. Big Spanish cities have received the largest population at risk. Although arrivals from endemic countries have decreased in the last years, we still observe many individuals at risk who have not been previously tested. Moreover, since univer-sal testing of pregnant women is not widely implemented, children with *T. cruzi* infection may be born and remain undiagnosed. In this study, individuals from endemic countries settled in Barcelona were mostly working-age women who came between 2004 and 2007. Migrants with a similar epidemiological profile, especially of Bolivian origin or descent, should be the target for a national or European screening program. Thus, health promotion programs and an ade-quate information to the patient in the consultation are essential to carry out the screening program.

In concurrence with other studies, previous contact with the vector was the most frequent and strongly associated risk factor with *T. cruzi* infection among those related with vector exposure.

The primary assessment in patients with chronic *T. cruzi* infection should include an ECG and echocardiogram. RBBB and LAFB were the most typical ECG disorders among *T. cruzi* infection patients in our series, while wall dilatation and diastolic dysfunction were the most frequent findings in the echocardiogram. In our series, as well as in other previous studies, a relevant proportion of patients with CC were diagnosed by echocardiogram hav-ing a normal ECG, which reinforces the echocardiogram as an essential test in the assess-ment of CC. Although dolichocolon was the most frequent finding in barium enema, it should not be interpreted as a digestive involvement of CD. It is expected that prevalence of CC and DCD will gradually increase in Europe as migrant populations settled in these countries age.

We are gradually learning more about *T. cruzi* infection, but there are still important knowledge gaps regarding this infection, especially about the lack of clinical or biological prog-nostic factors. Further studies will be necessary for a better understanding of its epidemiologi-cal and clinical behaviour.

## Supporting information

**S1 Table. The Supporting Information file contains the numbers used for the generation of Fig 1.** The file consists of five columns, one for each risk factor, and a sixth column indicating the number of subjects in the corresponding situation. For each risk factor, its presence or absence is indicated by TRUE or FALSE, respectively.
(CSV)

## Acknowledgments

We first would like to thank all the participants of this study. We also thank the Hospital Clínic de Barcelona and ISGlobal, as well as the nursing team, the medical team, and the staff who have helped in this project.

## Author Contributions

**Conceptualization:** Pedro Laynez-Roldán, Irene Losada-Galván, Elizabeth Posada, Leonardo de la Torre Ávila, Inés Oliveira-Souto, Antonia Calvo-Cano, José Muñoz, Joaquim Gascón, Maria-Jesus Pinazo.

**Data curation:** Pedro Laynez-Roldán, Irene Losada-Galván, Aina Casellas, Sergi Sanz, Carme Subirà, Isabel Vera, Montserrat Roldán, Edelweiss Aldasoro, Antonia Calvo-Cano, Maria-Eugenia Valls, Míriam J. Álvarez-Martínez, Alba Abras, Cristina Ballart, Maria-Jesus Pinazo.

**Formal analysis:** Pedro Laynez-Roldán, Irene Losada-Galván, Aina Casellas, Sergi Sanz, Maria-Jesus Pinazo.

**Funding acquisition:** Irene Losada-Galván, Joaquim Gascón, Maria-Jesus Pinazo.

**Investigation:** Pedro Laynez-Roldán, Irene Losada-Galván, Elizabeth Posada, Leonardo de la Torre Ávila, Aina Casellas, Sergi Sanz, Natalia Rodriguez-Valero, Daniel Camprubí-Ferrer, Isabel Vera, Montserrat Roldán, Inés Oliveira-Souto, Antonia Calvo-Cano, Maria-Eugenia Valls, Míriam J. Álvarez-Martínez, Montserrat Gállego, Alba Abras, Cristina Ballart, José Muñoz, Joaquim Gascón, Maria-Jesus Pinazo.

**Methodology:** Pedro Laynez-Roldán, Irene Losada-Galván, Aina Casellas, Sergi Sanz, Joaquim Gascón, Maria-Jesus Pinazo.

**Software:** Aina Casellas, Sergi Sanz, Carme Subirà.

**Supervision:** Irene Losada-Galván, Elizabeth Posada, Leonardo de la Torre Ávila, Natalia Rodriguez-Valero, Daniel Camprubí-Ferrer, Edelweiss Aldasoro, Inés Oliveira-Souto, Antonia Calvo-Cano, Montserrat Gállego, José Muñoz, Joaquim Gascón, Maria-Jesus Pinazo.

**Validation:** Joaquim Gascón, Maria-Jesus Pinazo.

**Visualization:** Pedro Laynez-Roldán, Irene Losada-Galván, Elizabeth Posada, Aina Casellas, Sergi Sanz, Carme Subirà.

**Writing – original draft:** Pedro Laynez-Roldán, Irene Losada-Galván, Joaquim Gascón, Maria-Jesus Pinazo.

**Writing – review & editing:** Pedro Laynez-Roldán, Irene Losada-Galván, Elizabeth Posada, Leonardo de la Torre Ávila, Aina Casellas, Sergi Sanz, Carme Subirà, Natalia Rodriguez-Valero, Daniel Camprubí-Ferrer, Isabel Vera, Montserrat Roldán, Edelweiss

Aldasoro, Inés Oliveira-Souto, Antonia Calvo-Cano, Maria-Eugenia Valls, Míriam J. Álvarez-Martínez, Montserrat Gállego, Alba Abras, Cristina Ballart, José Muñoz, Joaquim Gascón, Maria-Jesus Pinazo.

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
