## [Decision Letter · Decision Letter 0]

5 Jan 2023

Dear Mr Laynez-Roldán,

Thank you very much for submitting your manuscript "Characterization of Latin American migrants at risk for Trypanosoma cruzi infection in a non-endemic setting. Insights into initial evaluation of cardiac and digestive involvement" for consideration at PLOS Neglected Tropical Diseases. As with all papers reviewed by the journal, your manuscript was reviewed by members of the editorial board and by several independent reviewers. In light of the reviews (below this email), we would like to invite the resubmission of a significantly-revised version that takes into account the reviewers' comments. 

We cannot make any decision about publication until we have seen the revised manuscript and your response to the reviewers' comments. Your revised manuscript is also likely to be sent to reviewers for further evaluation.

[1] A letter containing a detailed list of your responses to the review comments and a description of the changes you have made in the manuscript [please make sure to indicate where changes have been made by specifying either paragraph or line numbers]. Please note while forming your response, if your article is accepted, you may have the opportunity to make the peer review history publicly available. The record will include editor decision letters (with reviews) and your responses to reviewer comments. If eligible, we will contact you to opt in or out.

Sincerely,

Richard Reithinger

Academic Editor

Ana Rodriguez

Section Editor

Reviewer's Responses to Questions

**Key Review Criteria Required for Acceptance?**

**Methods**

-Are the objectives of the study clearly articulated with a clear testable hypothesis stated?

-Is the study design appropriate to address the stated objectives?

-Is the population clearly described and appropriate for the hypothesis being tested?

-Is the sample size sufficient to ensure adequate power to address the hypothesis being tested?

-Were correct statistical analysis used to support conclusions?

-Are there concerns about ethical or regulatory requirements being met?

Reviewer #1: The authors present epidemiological and clinical data of Latin American immigrants to Spain who were found to have Trypanosoma cruzi infection at their hospital in Barcelona. 

-The objectives were to define risk factors for infection and describe clinical features of persons found to be infected. The objectives of this retrospective, descriptive study were straightforward and clear. As a descriptive study, hypotheses were not stated, but associations were presented, and their strength tested. 

-The study design was appropriate for this descriptive, retrospective study.

-The population was clearly described.

-The number of infected immigrants with T. cruzi infection identified and studied is perhaps the largest reported in the literature to date. The numbers were sufficient for robust analysis. 

-Standard and appropriate statistical analyses were performed

-There are no ethical concerns.

Reviewer #2: This manuscript does not tackle a specific research question.

**Results**

-Does the analysis presented match the analysis plan?

-Are the results clearly and completely presented?

-Are the figures (Tables, Images) of sufficient quality for clarity?

Reviewer #1: The analysis of data was straightforward and appropriate.

Data were presented clearly. However, it's not necessary to present hundredths (decimal points) in the percentages; round numbers or tenths of a percent are sufficient.

Results are clearly presented, but 2 small issues--table 2: OR said to be adjusted but not clear if ORs were adjusted or if adjusted, adjusted for what; OR for Bolivia is reversed

Not sure that map (figure 3) is necessary given highly selected study population

Reviewer #2: Tables are clear and well-structured in general. 

On the firgures, I think 

a) presenting a map may be misleadingly taken as real prevalence of T. cruzi infection (not the intention of the authors). 

b) A Venn diagram describing 5 factors includes too many numbers, making it difficult to the reader. I suggest to think in a different format to present this information.

**Conclusions**

-Are the conclusions supported by the data presented?

-Are the limitations of analysis clearly described?

-Do the authors discuss how these data can be helpful to advance our understanding of the topic under study?

-Is public health relevance addressed?

Reviewer #1: -Conclusions largely are supported by data with several exceptions. 

One is that all the ECG and echocardiographic abnormalities were evidence of Chagas heart disease. Given the nonspecific nature of many of the abnormalities, the authors should acknowledge that the estimates of Chagas heart disease may be overestimates. 

The second has to do with the echocardiogram. The authors point out that 10% of infected persons with a normal ECG had an abnormal echocardiogram, but 90% of these had diastolic dysfunction (a rate less than the general population!!); isolated diastolic dysfunction may not be an early manifestation of Chagas disease (this remains to be shown), and it may not predict progression, which is the hallmark of Chagas cardiomyopathy. Although current guidelines recommend an echocardiogram for initial evaluation, it should be clearly stated that techniques more sensitive than the ECG (such as the echocardiogram) will find minor abnormalities in high numbers of persons with T cruzi infection (see Am J Trop Med Hyg 2020;103:1480.). It has long been known the many people with "indeterminate" Chagas disease will have subtle abnormalities on more sensitive testing-the significance of these findings is unclear, but longitudinal data suggest strongly that most do not progress. The recommendation for doing echocardiograms in all persons with positive tests for T cruzi may have the unintended consequence of labelling someone as having classic Chagas cardiomyopathy (with its well-known bad long-term prognosis), when in fact the disease will never progress. The other downside of the guideline is that requiring an echocardiogram in all infected persons will provide yet another financial barrier for persons to undergo needed medical evaluation. 

-Major limitations of the study are pointed out by the authors. These primarily are a consequence of the highly selected study population (mostly Bolivian from a few areas, mostly women, many already diagnosed, many referred to study by doctor or because of disease). This limits the generalizability of study to other nonendemic countries.

-An important limitation of the study is lack of information on comorbidities (especially hypertension, coronary disease and risk factors, diabetes) . Comorbidities could account for the ECG abnormalities and interactions with Chagas disease will become increasingly important as the infected immigrant population ages.

-In terms of discussing how the data could be used to advance understanding of Chagas disease in immigrant populations, the authors should mention the importance of long-term follow-up of their cohort to answer some of the questions about progression of disease outside of endemic areas.

-The public health relevance of the study speaks for itself and is appropriately emphasized by the authors

Reviewer #2: Yes, in general, authors conclusions are based on their reported data.

**Editorial and Data Presentation Modifications?**

Reviewer #1: Although there is no major problem with the overview of Chagas disease, it is extremely long and may distract from the major point of the paper, which is to describe experience in a nonendemic country. This will be up to the Journal editors ultimately, but we would strongly recommend decreasing the introduction to a few paragraphs in order to focus on the important parts of the paper. 

There are some small items in the introduction that need attention if it is retained. Some of the statements may be outdated as noted. Numbers refer to line number in the text

Line 92: Does in fact traditional vectorborne transmission still account for most new infections? It does account for most existing infections.

 Line 93-4: Metacyclic tryptomastigotes do not penetrate the blood through the skin

94. T infestans was the most important vector—is it still? 

96. Rather than say oral transmission is possible would point out its importance as a route of transmission. Over 5000 cases of oral transmission have been reported and in many areas this route of transmission far outweighs traditional vector borne transmission. 

104-105 Rates of transfusion transmitted infection depend on the blood product being transfused; the 10-20% data are old and probably reflect transmission of whole blood rather than individual products; currently, platelets are the most common vehicle; risk of transfusion-induced transmission may vary among geographic areas and presence or absence of ongoing vectorborne transmission. 

 106-7. Would clarify that transmission in North America is in the United States and not Canada; Mexico is part of North America but is high rates of active borne transmission. 

110. "Orography” pertains to mountains primarily and although relevant to Bolivia, may be confusing to readers. 

Line 125: Infection in the GI tract most significantly involves the autonomic ganglia. 

Line 144. Would say the DNA amplification not useful for diagnosis of chronic infection rather than saying it has little relevance in the chronic phase as pointed out in 149—this may be confusing to readers as written

In the Discussion, first paragraph, a more extensive review of studies outside of Europe would be appropriate for this paper, which focuses on experiences in nonendemic countries. For instance, see Curr Trop Med Rep 2022, Sept 10:1-9 (available through pub med) for summary of US studies.

Reviewer #2: 1. The text equates Trypanosoma cruzi infection with Chagas diseases. I think this is not correct, as the later term refers to the fraction of the population developing abnormalities. 

2. The intro section (line 150-1529) states that a 12-lead ECG should be taken to all seropositive individuals. However, the data presented in this report suggest it lacks sensitivity to detect cardiac abnormalities. This fact should be addressed in the discussion

3. The statement "anti-Trypanosoma therapy could reduce disease progression.." (lines 389-390) may be unsubstantiated. The authors selectively cite a published observational study, excluding two randomized trials, with opposite results (one of them unpublished, but presented as abstract in at least one medical meeting).

**Summary and General Comments**

Reviewer #1: The authors present their experience over the last 17 years evaluating immigrants from Latin America for Chagas disease; the group has the largest experience in nonendemic countries, and for this reason the current paper is very welcome.

The paper could be improved by condensing and focusing the introduction, and tightening up the discussion, which is lengthy. If this paper is meant to be followed by a long-term follow-up study (hopefully so!) the authors might mention this. They might also mention the importance of comorbidities and aging on the outcome of T cruzi infection.

Reviewer #2: This is an informative, well-organized, well-written manuscript, based on two decades of clinical assessments to immigrants from Latin-American countries, potentially carrying T. cruzi infection. The authors deserve all credit and should be congratulated for completing this work over the years. 

I found, however, difficulty reading it as an original research article. The paper describes different aspects of patients attending a tropical medicine clinic, after using conventional methods for their diagnosis and clinical assessment. It does not entail a specific research question. Further, with an unusually long introduction, covering mostly general aspects of Chagas disease, the text reads initially as a narrative review/tutorial. 

Their main findings, that Bolivian immigrants made 85% of the participants, that 48% were seropositive, and that their clinical characteristics were (as expected) similar as those of infected individuals living elsewhere can hardly be seen as novel. There would also be validity concerns, if we were to see this as a survey/registry of infected patients, mainly because of selection bias. Firstly, as a sample of the infected population: this is a single center study, including mostly asymptomatic patients from a region in Bolivia (the fifth country of origin of Latin American immigrants to Spain) attending the clinics for different reasons. Secondly, as assessment of the study population: only half of the participants had an ECG taken, and 12.8% had an echocardiogram, not necessarily based on prior ECG findings. 

I think this report raises awareness about Chagas disease as a global health problem. However, in its current format, I cannot see this report as a research manuscript. Authors may want to elaborate more specific questions, taking advantage of the information at hand. Alternatively, this may be re-arranged as a narrative/tutorial review that is illustrated with the experience/data.

PLOS authors have the option to publish the peer review history of their article (what does this mean?). If published, this will include your full peer review and any attached files.

Reviewer #1: No

Reviewer #2: Yes: Juan Carlos Villar
---

## [Editor Report · Decision Letter 1]

27 Mar 2023

Dear Mr Laynez-Roldán,

Thank you very much for submitting your manuscript "Characterization of Latin American migrants at risk for Trypanosoma cruzi infection in a non-endemic setting. Insights into initial evaluation of cardiac and digestive involvement" for consideration at PLOS Neglected Tropical Diseases. As with all papers reviewed by the journal, your manuscript was reviewed by members of the editorial board and by several independent reviewers. The reviewers appreciated the attention to an important topic. Based on the reviews, we are likely to accept this manuscript for publication, providing that you modify the manuscript according to the review recommendations. 

Thank you for addressing all of the reviewers' concerns. I did think that the manuscript warranted some small re-structuring and tightening of the language. Please see my edits in the attached file for proposed changes. Let me know if these are agreeable to you.

Sincerely,

Ana Rodriguez

Section Editor

Ana Rodriguez

Section Editor

Thank you for addressing all of the reviewers' concerns. I did think that the manuscript warranted some small re-structuring and tightening of the language. Please see my edits in the attached file for proposed changes. Let me know if these are agreeable to you.

Figure Files:

Data Requirements:

Reproducibility:

References

---

## [Editor Report · Decision Letter 2]

24 Apr 2023

Dear Mr Laynez-Roldán,

We are pleased to inform you that your manuscript 'Characterization of Latin American migrants at risk for Trypanosoma cruzi infection in a non-endemic setting. Insights into initial evaluation of cardiac and digestive involvement' has been provisionally accepted for publication in PLOS Neglected Tropical Diseases.

Best regards,

Richard Reithinger

Academic Editor

Ana Rodriguez

Section Editor

Thank you for accepting all of the suggested edits and changes.

---

## [Editor Report · Acceptance letter]

14 Jun 2023

Dear Mr Laynez-Roldán,

We are delighted to inform you that your manuscript, "Characterization of Latin American migrants at risk for Trypanosoma cruzi infection in a non-endemic setting. Insights into initial evaluation of cardiac and digestive involvement," has been formally accepted for publication in PLOS Neglected Tropical Diseases.

Best regards,

Shaden Kamhawi

co-Editor-in-Chief

Paul Brindley

co-Editor-in-Chief
